# Clinical Utility of Comprehensive Genomic Profiling in Patients with Unresectable Hepatocellular Carcinoma

**DOI:** 10.3390/cancers15030719

**Published:** 2023-01-24

**Authors:** Shun Ishido, Kaoru Tsuchiya, Yoshihito Kano, Yutaka Yasui, Kenta Takaura, Naoki Uchihara, Keito Suzuki, Yuki Tanaka, Haruka Miyamoto, Michiko Yamada, Hiroaki Matsumoto, Tsubasa Nobusawa, Taisei Keitoku, Shohei Tanaka, Chiaki Maeyashiki, Nobuharu Tamaki, Yuka Takahashi, Hiroyuki Nakanishi, Urara Sakurai, Yasuhiro Asahina, Ryuichi Okamoto, Masayuki Kurosaki, Namiki Izumi

**Affiliations:** 1Department of Gastroenterology and Hepatology, Musashino Red Cross Hospital, Tokyo 180-8610, Japan; 2Department of Gastroenterology and Hepatology, Tokyo Medical and Dental University, Tokyo 113-8510, Japan; 3Department of Clinical Oncology, Tokyo Medical and Dental University, Tokyo 113-8510, Japan; 4Department of Pathology, Musashino Red Cross Hospital, Tokyo 180-8610, Japan

**Keywords:** hepatocellular carcinoma, comprehensive genomic profiling, systemic therapy, individualized therapy, molecular mechanism

## Abstract

**Simple Summary:**

This study aimed to investigate the clinical usefulness of comprehensive genomic profiling (CGP) in patients with unresectable hepatocellular carcinoma who received multiple systemic therapies in real-world practice. In this study, all nine patients had gene alterations, and seven were candidates eligible for clinical trials based on the results of CGP. The median number of alterations per patient was four, and the blood sample was used in five patients with extrahepatic metastasis. We revealed the genomic information of the patients who received multiple systemic therapies and reported the utility of blood samples in patients with extrahepatic metastasis. Furthermore, the genomic status in patients treated with multiple molecular-targeted agents, including checkpoint inhibitors, would contribute to developing newer systemic agents.

**Abstract:**

The molecular mechanism of hepatocellular carcinoma (HCC) is partially demonstrated. Moreover, in the patients receiving multiple molecular-targeted therapies, the gene alternations are still unknown. Six molecular-targeted therapies of unresectable HCC (uHCC) and comprehensive genomic profiling (CGP) have been approved in clinical practice. Hence, the utility of CGP in patients with uHCC treated with multiple molecular-targeted agents is investigated. The data of the patients with uHCC who received CGP tests were collected, retrospectively, between February 2021 and May 2022. Gene alterations detected by foundation testing, excluding variants of unknown significance, were reported in all nine patients. The samples for CGP were derived from liver tumor biopsy (*n* = 2), surgical specimens of bone metastases (*n* = 2), and blood (*n* = 5). The median number of systemic therapies was four. Seven patients were candidates eligible for clinical trials. One patient with a high tumor mutation burden (TMB) could receive pembrolizumab after CGP. This study presented genomic alternations after receiving multiple molecular-targeted therapies. However, further investigation needs to be conducted to develop personalized therapies and invent newer agents for treating HCC.

## 1. Introduction

Liver cancer is a global health problem and has become the second and sixth most common cause of cancer-associated deaths in men and women, respectively, in 2020 [1]. Hepatocellular carcinoma (HCC) accounts for 75–85% of liver cancer cases. The life expectancy of patients with HCC has improved considerably with rapid advancement in systemic therapy, including immunotherapy. However, based on recent clinical trials, the median overall survival of advanced HCC is 15–19 months [2,3,4]. Moreover, a few biomarkers were used for decision-making. The features of the molecular pathogenesis and drivers of HCC have already been reported partially [5,6,7]. However, most previous studies included only patients who had undergone resection or received a few systemic therapies. In patients with unresectable HCC (uHCC), six systemic therapies, including sorafenib [8,9], regorafenib [10], Lenvatinib [2], ramucirumab (only AFP ≥ 400 ng/mL) [11], cabozantinib [12], and atezolizumab plus bevacizumab [3], have been reimbursed by national health insurance in Japan. Moreover, clinical sequencing in tissue specimens has been covered by national health insurance since June 2019. Subsequently, a liquid biopsy was also approved in August 2021 [13,14]. This has led to a rapidly expanding number of individualized therapies that specifically target comprehensive genomic profiling in a patient’s tumor. Personalized cancer therapy could be achieved by regulating the mutation status of specific molecular drivers in critical signaling pathways. However, its clinical utility in uHCC remains unknown. In this study, the clinical utility of comprehensive genomic profiling with Foundation One^®^ CDx (F1CDx) and Foundation One^®^ Liquid CDx (F1LCDx) in real-world practice is investigated. These systems are used to identify potentially actionable genetic alterations and perform precision individualized therapies. Furthermore, the genomic status in patients with HCC after receiving multiple systemic therapies in real-world practice would be revealed in the future to develop newer molecular-targeted agents.

## 2. Materials and Methods

### 2.1. Study Protocol

The patients who had uHCC and underwent F1CDx or F1LCDx at our hospital between February 2021 and May 2022 were, retrospectively, investigated. All the patients either progressed or were finishing the standard systemic therapy. Before enrolling for the study, written informed consent was obtained from each patient. The study protocol conformed to the ethical guidelines of the Declaration of Helsinki, and the study was approved by the institutional ethics review committee (approval number: 2102). The sequencing test, data analysis, and annotation were conducted by Foundation Medicine Inc. The final report on F1CDx or F1LCDx included any detected genomic findings and FDA-approved therapeutic options.

### 2.2. Clinical and Laboratory Data

The clinical data, including the age, sex, performance status, liver function, renal function, nutritional status, tumor markers (AFP and PIVKA-II), and imaging findings from the initial systemic treatment for HCC to the last visit after the comprehensive genomic profiling test were collected. Each systemic treatment was performed as per the manufacturer’s guidelines. Dynamic computed tomography (CT) was performed at baseline and, thereafter, every 6–12 weeks. Dynamic magnetic resonance imaging (MRI) was conducted for patients allergic to the contrast agents of the CT scan. The treatment response was evaluated based on the Response Evaluation Criteria in Solid Tumors (RECIST ver1.1) or Modified Response Evaluation Criteria in Solid Tumors (mRECIST). Adverse events (AEs) were reported as per the Common Terminology Criteria for Adverse Events (CTCAE) version 5.0.

### 2.3. F1CDx and F1LCDx

F1CDx and F1LCDx are qualitative next-generation sequencing that use targeted high-throughput hybridization-based capture technology for the detection of substitutions, insertion and deletion alterations, and copy number alterations (CNAs) in 324 genes and select gene rearrangements, using DNA isolated from formalin-fixed, paraffin-embedded (FFPE) tumor tissue specimens or blood. These tests are intended to identify patients who may benefit from treatment with therapies in accordance with approved therapeutic product labeling. Foundation One^®^ Liquid CDx is an FDA-approved companion diagnostic that analyzes guideline-recommended genes from a simple blood draw. It is the only FDA-approved blood-based test to analyze over 300 genes—making it the most comprehensive FDA-approved liquid biopsy on the market. To detect base substitution, reads with low mapping (mapping quality < 25) or base calling quality (base calls with quality ≤ 2) were discarded. Final calls were made at mutant allele frequency (MAF) ≥ 5% (MAF ≥ 1% at hotspots). Variants were classified as variants of uncertain (VUS) when the significance and impact upon cancer progression were unknown due to a lack of reported evidence and conclusive change in function based on the previous reports [15,16]. Genomic DNA control samples were analyzed at a central laboratory testing service.

### 2.4. Expert Panel Discussion

After F1CDx or F1LCDx, each case was reviewed at the expert panel discussion with specialists, including genetic counselors, pathologists, medical oncologists, bioinformaticians, clinical research coordinators, and primary physicians. Based on the patient’s medical treatment or family history, the genetic results of actionable genomic alterations and treatment options were carefully evaluated by these specialists.

### 2.5. Treatment after F1CDX or F1LCDX

After the expert panel discussion, each doctor explained the results to the patients. Additionally, if relevant clinical trials were available, the patients were introduced to the National Cancer Center. If relevant clinical trials were not available, the therapeutic strategies were decided based on the discussion with the tumor board.

## 3. Results

### 3.1. Patient Characteristics

During the study period, nine patients received F1CDX or F1LCDx. The characteristics of the patients with F1CDX or F1LCDx are listed in Table 1. The patients defined as others were without hepatitis B virus (HBV), hepatitis C virus (HCV), or alcohol consumption. The mean age of patients defined as others was 30 years old and younger than those with HBV, HCV, or alcohol consumption. Four patients had a smoking history (current or former smokers). One patient with HBV was treated for esophageal varices by endoscopic variceal ligation (EVL) before CGP. Three patients (two patients with HBV and one patient with HCV) had low platelets (<10⁵/µL) at the CGP test. All patients had no cancer other than hepatocellular carcinoma.

Extrahepatic metastasis was observed in seven patients; peritoneal dissemination (*n* = 1), lung only (*n* = 3), and lung and bone (*n* = 3). The previous systemic therapies in each patient are shown in Table 2. The number of systemic therapies received was six in one patient, five in three patients, and four in five patients. The samples for F1CDX or F1LCDx were from liver tumor biopsy (*n* = 2), surgical specimens of bone metastases (*n* = 2), and blood (*n* = 5). All patients who received F1LCDx had extrahepatic metastasis, while two of four patients evaluated by F1CDx had only intrahepatic lesions, including major vascular invasion. Liver function was maintained as Child–Pugh A in three of four patients with F1CDx, while two of five patients were diagnosed as Child–Pugh B at the CGP test.

### 3.2. Common Alterations in Patients with HCC

The most commonly altered genes, excluding VUS in HCC, are shown in Figure 1. There are alterations in all patients with HCC. The two most frequent alterations observed were TP53 (*n* = 3, 33.3%) and *CTNNB1* (*n* = 3, 33.3%). Four patients were investigated by using F1CDx, and the other five patients were investigated by using F1LCDx (Table 1). Five patients performed F1LCDx, and detectable alterations were seen in all patients. Based on the results of comprehensive genomic profiling, four out of five patients were candidates for the clinical trials. In Appendix A, Table 2 and Appendix A, the genomic results of nine patients are shown. All patients had detectable alterations, and the median number of alterations (except VUS) per patient was four (2–5). The results of tumor mutation burden (TMB) and microsatellite instability (MSI) were obtained. The median number of TMB was four (range 0–20). Only one patient had TMB-high (TMB ≥ 10 mutations/megabase), whereas none of the patients had MSI-high (MSI sensor scores ≥ 10). An alternation suspected of a hereditary tumor was not seen in any patient. The results of comprehensive genomic profiling in patients with uHCC are illustrated in Figure 2. Five patients performed F1LCDx, and detectable alterations were seen in all patients. All patients evaluated with F1LCDx had extrahepatic metastasis. One patient received six regimens, and four of five patients experienced four regimens before CGP. Based on the results of comprehensive genomic profiling, four out of five patients were candidates for the clinical trials.

### 3.3. Treatment after The Comprehensive Genomic Profiling

The clinical outcome after CGP is described in Table 3. Based on the results of the comprehensive genomic profiling, the patient with TMB-high was treated with pembrolizumab, whereas the other patient was treated with regorafenib. This decision was taken based on the profiling outcome by the expert panel, which revealed that regorafenib would be a more effective option. This patient was retreated with regorafenib and survived for five months after the retreatment. The OS from the initial liver resection was 12.5 years, and survival from the initiation of sorafenib treatment for lung metastases was 4.5 years. One patient with a *CTNNB1* mutation received cabozantinib as the fifth line of treatment and maintained SD for 11 months. Based on the result of gene profiling by multigene panel test, the other patient with *TSC2* planned to participate in everolimus’s prospective trial of patient-proposed healthcare services with multiple targeted agents. Of these nine patients, seven (77.8%) were eligible for participating in ≥1 clinical trial option. However, none could enter the clinical trials. The reasons for the inability of patients to participate in the trials are listed in Table 3. Of the nine patients, three could not participate due to renal dysfunction or low platelet counts, and one due to HBV infection. The median survival duration from the comprehensive genomic profiling was 4 (2–18) months.

## 4. Discussion

Based on our knowledge, this is the first report to reveal the clinical utility of comprehensive genomic profiling in patients with uHCC after receiving multiple molecular-targeted therapies. Llovet et al. [5] reviewed the molecular pathogenesis of HCC and revealed that most molecular alternations were undruggable, with only 20–25% of tumors having an actionable driver mutation. However, all nine patients in this study had alterations, and the median number of alterations per patient was four. Of the nine patients, seven (77.8%) were eligible for participating in the ≥1 clinical trial, which was conducted as per the comprehensive genomic profiling in malignant tumors. The results of this study differed from past studies because of the different backgrounds of the patients. In this study, all patients received atezolizumab plus bevacizumab and received more than three molecular-targeted therapies. The patients who received sequential therapies for HCC had actionable driver mutations more frequently compared to those registered in previous studies. In recent studies [5,6,7], *CTNNB1* mutations were observed in 33% of patients. *CTTNB1* mutation is associated with activation of Wnt/β-catenin signaling and enriched in non-T-cell inflamed HCCs, which demonstrated a poor clinical response to an immune checkpoint inhibitor. Ogawa K et al. [17] reported that in patients with HCC, the treatment effect of atezolizumab plus bevacizumab with mutant *CTNNB1* was comparable to the patients with wild-type *CTNNB1*. In this study, of the nine patients, three had a *CTNNB1* mutation. The best responses during atezolizumab plus bevacizumab were stable disease (SD) in two patients and progressive disease (PD) in one patient, as per RECIST ver1.1.

One of the most important findings of this study was revealing the utility of liquid biopsy in patients with uHCC. Of the five patients performing F1LCDx, detectable alterations were seen in all patients. Based on the results of comprehensive genomic profiling, four out of five patients were candidates for the clinical trials. Translating molecular knowledge into precision oncology has been perceived as difficult in HCC because of the existence of intratumoral heterogeneity and multifocal tumors. Under such circumstances, a liquid biopsy, which is a noninvasive technique with the demonstration of the genetic information representative of the tumor genome, could be highly valuable.

Although this study included a small number of patients, some patients had significant mutations associated with β-catenin inhibitors and mTOR inhibitors. In this study, of the nine patients, three had *CTNNB1* mutation, which is associated with β-catenin degradation. Subsequently, *CTNNB1* mutation leads to the constitutive activation of Wnt/β-catenin signaling. An orally active selective inhibitor of the interaction between β-catenin and CREB binding protein, E7386 [18], demonstrated antitumor activity through the modulation of the Wnt/β-catenin signaling pathway. In Japan, the clinical trial of combination therapy with E7386 and PD-1 antibody (ClinicalTrials.gov Identifier: NCT05091346) is ongoing. The other anticipated agent is the mTOR inhibitor. In HCC, the PI3K/AKT/mTOR signaling pathway was reported, and its activation was frequently detected [19,20]. Chen J.S. et al. [21] reported that the PI3K/PTEN/AKT/mTOR pathway was involved in invasion and metastasis in HCC. Ocana A. et al. [22] revealed that the activation of the PI3K/mTOR/AKT pathway was associated with significantly worse 5-year survival in solid tumors. In this study, molecular aberrations led to the putative activation of the mTOR pathway and were detected in five of the nine patients. In advanced HCC, a preliminary antitumor effect was reported in a phase I/II study of mTOR inhibition using the rapamycin derivate RAD001 (everolimus) [23]. However, in a subsequent phase III trial (EVOLVE-1), everolimus did not meet the primary endpoint [24]. The patients who participated in this study did not receive sequential therapies and were treated only with sorafenib.

All nine patients in this study had gene alternations, and one patient received pembrolizumab based on the results of CGP. The other patient participated in the clinical trial for unresectable HCC instead of the everolimus’s prospective trial of patient-proposed healthcare services with multiple targeted agents. Three patients visited the National Cancer Center Hospital to participate in the clinical trials based on CGP. However, they were excluded because of renal dysfunction, low platelet count, or HBV infection. In Japan, comprehensive genomic profiling tests are approved only in patients who have finished the standard therapies or have received the last recommended treatments. Furthermore, the timing of the comprehensive genomic profiling tests should be discussed at a multidisciplinary tumor board.

This study has a few limitations. First, it was a single-centric study. Hence, the sample size was very small. We could not perform any statistical analysis. Second, all the participants of this study were Japanese. Hence, ethnicity could not be discussed. In the future, further research should be conducted with international multicentric investigations, including a large sample size to confirm the results of this study. Third, this study was conducted in Japan. However, the reimbursement laws differ in each country. Further studies, including a large number of patients in multi centers, are necessary.

This study is the first to reveal the usefulness of comprehensive genomic profiling in patients with uHCC receiving multiple molecular-targeted therapies. Although no patient could participate in the clinical trials based on CGP, the reasons were associated with organ function or viral infection and not gene alternations. The results of this study contribute to a better understanding of the genomic status and development of newer molecular-targeted agents for HCC.

## 5. Conclusions

Comprehensive genomic profiling with tumor tissue or patient’s blood might be useful in patients with HCC receiving sequential molecular-targeted therapies. In the future, the best timing for comprehensive genomic profiling should be discussed to provide more personalized treatment to the patient.

## Figures and Tables

**Figure 1 cancers-15-00719-f001:**
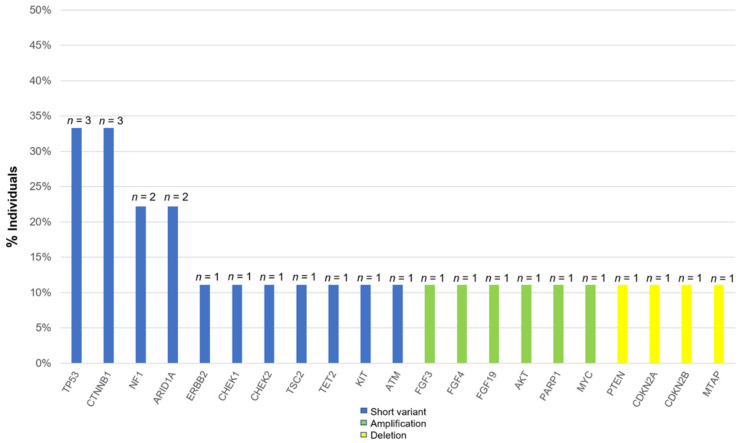
Percentage of all alterations in patients with unresectable hepatocellular carcinoma.

**Figure 2 cancers-15-00719-f002:**
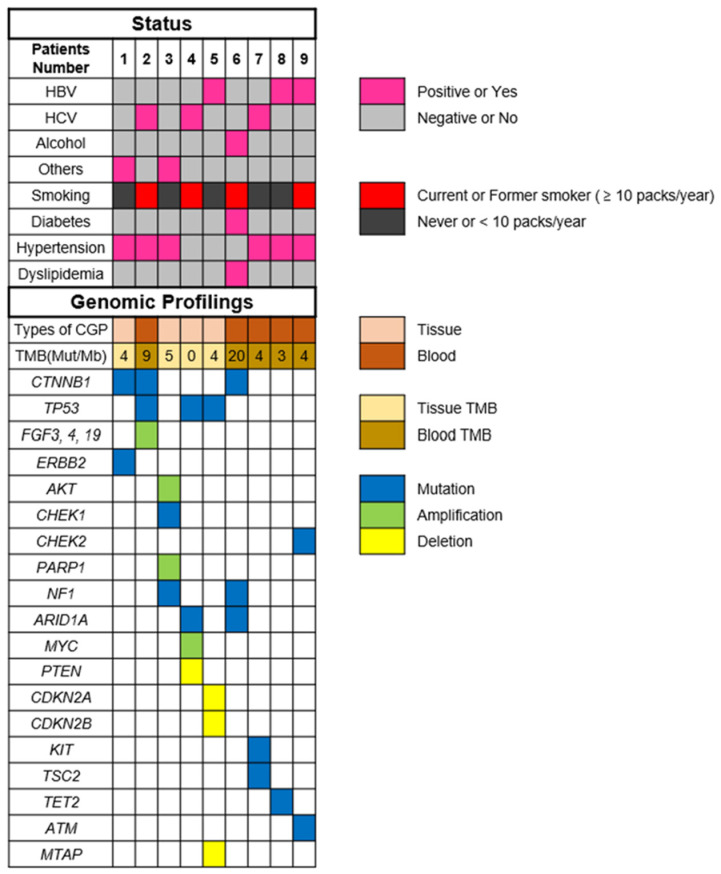
Results of comprehensive genomic profiling in patients with unresectable hepatocellular carcinoma. Abbreviations: TMB—tumor mutation burden; HBV—hepatitis B virus; HCV—hepatitis C virus; and others mean non-HBV, non-HCV, and non-alcohol.

**Table 1 cancers-15-00719-t001:** Baseline characteristics of the patients at comprehensive genomic profiling test.

*n* = 9	
**Age (Years), Median (IQR)**	65 (25–78)
**Sex: Male/Female (%)**	7 (77.8%)/2 (22.2%)
**Body Weight (kg), Median (IQR)**	57.7 (47.6–102.5)
**Etiology HBV/HCV/Alcohol/Others (%)**	3 (33.3%)/3 (33.3%)/1 (11.1%)/2 (22.2%)
**Platelets (10⁴/μL), Median (IQR)**	16.2 (6.2–31.4)
**AST (U/L), Median (IQR)**	42 (22–65)
**ALT (U/L), Median (IQR)**	34 (13–96)
**Total Bilirubin (mg/dL), Median (IQR)**	0.6 (0.4–2.6)
**Albumin (g/dL), Median (IQR)**	3.6 (2.2–4.0)
**Prothrombin Time–International Normalized Ratio, Median (IQR)**	1.0 (0.9–1.2)
**Creatinine (mg/dL), Median (IQR)**	1.0 (0.5–1.4)
**eGFR (mL/min/1.73^m2^), Median (IQR)**	58.8 (42.1–149.3)
**Urine Total Protein/Creatinine**	0.4 (0.1–7.9)
**Child–Pugh A/B/C (%)**	6 (66.7%)/3 (33.3%)/0
**ECOG PS 0/1/2 (%)**	8 (88.9%)/1 (11.1%)/0
**BCLC Stage A/B/C (%)**	1 (11.1%)/0/8 (88.9%)
**Macroscopic Vascular Invasion, Yes/No (%)**	2 (22.2%)/7 (77.8%)
**Extrahepatic Metastasis, Yes/No (%)**	7 (77.8%)/2 (22.2%)
**Baseline AFP Concentration (ng/mL),** **Median (IQR)**	6519.4 (1.7–19827.3)
**Baseline AFP < 400 ng/mL, Yes/No (%)**	4 (44.4%)/5 (55.6%)
**Baseline DCP Concentration (mAU/mL),** **Median (Range)**	2965.5 (14.9–36091.5)
**Number of Past Systemic Therapies** **1/2/3/4/5/6 (%)**	0/0/0/5 (55.6%)/3 (33.3%)/1 (11.1%)
**Specimen Collection** **Liver/Bone/Blood**	2 (22.2%)/2 (22.2%)/5 (55.6%)
**Types of CGP** **Foundation One^®^ CDx** **Foundation One^®^ Liquid CDx**	4 (44.4%) 5 (55.6%)
**Initial HCC Treatment** **Resection/RFA/TACE/Systemic Therapy (%)**	4 (44.5%)/2 (22.2%)/0/3 (33.3%)
**Observation Period from the Initial HCC Treatment (Months), Median (Range)**	57 (13–144)

Note: Data are shown as the median number. Abbreviations: HBV—hepatitis B virus; HCV—hepatitis C virus; AST—aspartate aminotransferase; ALT—alanine aminotransferase; eGFR—estimated glomerular filtration rate; ECOG—electrocochleography; BCLC—Barcelona clinic liver cancer; AFP—alpha fetoprotein; DCP—des-gamma-carboxy prothrombin; HCC—hepatocellular carcinoma; RFA—radiofrequency ablation; TACE—transcatheter arterial chemoembolization; and CGP, comprehensive genomic profiling. Foundation One^®^ CDx (F1CDx) performs CGP using tissue specimens, while Foundation OneⓇ Liquid CDx (F1LCDx) performs CGP using blood.

**Table 2 cancers-15-00719-t002:** The results of comprehensive genomic profiling test.

Patients Number, Age, Sex	Etiology	Number of Prior Systemic Therapies	Metastatic Spread	CGP Method (Organ)	Mutations (VAF%)
① 20s male	NBNC	4 LEN + TAI→ATZ + BEV→SOR→REG	Nothing	F1CDx (liver)	*CTNNB1 G34R* (18.9%)/ *D32Y* (6.2%) *ERBB2 P967Q* (50%)
② 70s female	HCV	6 SOR→REG→LEN→ATZ + BEV→RAM→CAB	Lungs, bones	F1LCDx (blood)	*CTNNB1 S33C* (0.15%) *FGF3* (present*) *FGF4* (present*) *FGF19* (present*) *TP53* (0.53%)
③ 30s male	NBNC	5 SOR→REG→LEN→RAM→ATZ + BEV	Lungs, bones	F1CDx (bone)	*AKT3* (amplification) *CHEK1 K224** (54.6%) *PARP1* (amplification) *NF1 R2083C* (present*)
④ 70s male	HCV	4 LEN→ATZ + BEV→CAB→SOR	Lungs, bones	F1CDx (bone)	*TP53 S227fs*2* (46.9%) *PTEN* (loss exons) *ARID1A* splice site *2988 + 2T > G* (62%) *MYC* (amplification)
⑤ 50s male	HBV	5 LEN→HAIC→ATZ + BEV→RAM→CAB	Nothing	F1CDx (liver)	*CDKN2A/B* (loss exsons) *MTAP* (loss exsons) *TP53 R249S* (80.2%)
⑥ 60s male	ALD	4 LEN + TAI→ATZ + BEV→SOR→REG	Lungs	F1LCDx (blood)	*TMB-high* (20 mutations/megabase) *NF1 Y80fs*26* (1.4%) *ARID1A K990fs*18* (18%) *CTNNB1 K335T* (16.4%)
⑦ 60s male	HCV	4 LEN→ATZ + BEV→CAB→REG	Lungs	F1LCDx (blood)	*KIT D816E* (0.29%) *TSC2* splice site *2587_2639 + 36del89* (0.42%)
⑧ 60s male	HBV	4 LEN→ATZ + BEV→CAB→RAM	Lungs	F1LCDx (blood)	*TET2 H1904R* (0.55%)
⑨ 60s male	HBV	5 SOR→REG→ATZ + BEV→LEN→CAB	Peritoneal dissemination	F1LCDx (blood)	*CHEK2 T43fs*15* (0.15%) *ATM R1068fs*18* (1.5%)

Abbreviations: NBNC—non-HBV and non-HCV; HBV—hepatitis B virus; HCV—hepatitis C virus; ALD—alcoholic liver disease; SOR—sorafenib; REG—regorafenib; LEN—Lenvatinib; TAI—transcatheter arterial infusion; ATZ + BEV—atezolizumab plus bevacizumab; RAM—ramucirumab; CAB—cabozantinib; HAIC—hepatic arterial infusion chemotherapy (low-dose fluorouracil plus cisplatin); FGFR—fibroblast growth factor receptors; PD-1—programmed cell death 1; MEK—mitogen-activated extracellular-signal-regulated kinase; mTOR—mammalian target of rapamycin; ATRA—ataxia telangiectasia and Rad3-related protein; F1LCDx—Foundation One^®^ CDx; and F1LCDx—Foundation One^®^ Liquid CDx. VAF% means variant allele frequency percentage. Present* means VAF% is not applicable.

**Table 3 cancers-15-00719-t003:** Clinical outcome after comprehensive genomic profiling test.

Patients Number, Age, Sex	Reasons for Not Participating in Clinical Trials	After CGP	Survival Durationafter CGP
① 20s male	Excluded from the clinical trial because of renal dysfunction	Treated with CAB (based on CGP) because CAB was reported to inhibit the beta-catenin pathway partially	18 months
② 70s female	irAE	Treated with REG (based on CGP). The expert panel recommended that retreatment of regorafenib would be an effective option	6 months
③ 30s male	Excluded from the clinical trial because of low platelet counts	Treated with CAB as 6th line	17 months
④ 70s male	Worsening PS	Treated with RAM as 5th line	5 months
⑤ 50s male	No applicable clinical trials	BSC	4 months
⑥ 60s male	The patient decided to receive pembrolizumab with health insurance by national health insurance	Treated with pembrolizumab (based on CGP) because the CGP found TMB-high	3 months
⑦ 60s male	Planned to participate in everolimus’s prospective trial of patient-proposed healthcare services with multiple targeted agents	The patient participated in the other clinical trial at National Cancer Center	2 months
⑧ 60s male	No applicable clinical trials	Treated with SOR as 5th line	2 months
⑨ 60s male	Excluded from the clinical trial because of HBV infection	Retreatment with ATZ + BEV	3 months

Abbreviations: irAE—immune-related adverse events; PS—performance status; PD-1—programmed cell death 1; HBV—hepatitis B virus; CGP—comprehensive genomic profiling; TMB-high—high tumor mutation burden (TMB ≥ 10 mutations/megabase); CAB—cabozantinib; REG—regorafenib; RAM—ramucirumab; BSC—best supportive care; SOR—sorafenib; and ATZ + BEV—atezolizumab plus bevacizumab.

## Data Availability

All data generated or analyzed during this study are included in this article. Further inquiries can be directed to the corresponding author.

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
