# Peer review of "Clinical Utility of Comprehensive Genomic Profiling in Patients with Unresectable Hepatocellular Carcinoma"

_cancers, 2023, doi:10.3390/cancers15030719_

Round 1

Reviewer 1 Report

Ishido S. and collaborators investigated the clinical usefulness of comprehensive genomic profiling (CGP) in patients with unresectable hepatocellular carcinoma (uHCC) who received multiple systemic therapies in real-world practice. While Hepatocellular carcinoma (HCC) accounts for 75%–85% of primary liver cancer cases, the life expectancy of these patients has considerably improved with rapid advancement in systemic therapy. Recently, clinical trials have shown that median overall survival of advanced HCC was months, and only few biomarkers were used for decision-making. Moreover, national health insurance in Japan have been covering six systemic therapies, and sequencing of solid/liquid samples from patients with uHCC. This has led to a rapidly expanding number of individualized therapies that specifically target CGP in a patient's tumor. However, the clinical utility of these tools in uHCC remains unknown, for these reasons the authors aimed to investigate the clinical utility of CGP by in real-world practice by using FoundationOne®CDx (F1CDx) and FoundationOne®Liquid CDx (F1LCDx).

This work raised some questions and comments:

1.     The authors mention within the Abstract, in Background (line 27-28), “The major part of the molecular mechanism of hepatocellular carcinoma 27 (HCC) is already demonstrated”. This claim it’s not entirely true, because there are HCC subtypes around the world associated to non-cirrhotic livers which the main cause and molecular mechanisms remain unknown. The same claim is also found in Introduction (line 52-53). The authors focus on the problematics of developed countries (as in Japan), they should not generalize their claims because as mentioned above it’s not entirely true worldwide.

2.     Results

a.     The authors talk about the etiologies of the evaluated patients (Table 1), however this information it’s not used at all in the following results. It should be interesting to know, for example if patients infected with Hepatitis B virus are younger than those with Hepatitis C virus infection or alcohol consumption.

b.     The authors also stress the fact that they used 2 types of tissue collection methods (F1CDx and F1LCDx). Could be interesting and more informative to also add to Table 1, 2 columns with the characteristics of the patients by tissue collection method. This information could also allow to better understand the data structure and if there are some bias associated to the way of collection.

c.     Table 2 should be merged with Table 1, and this will be possible by what was recommended above (add tissue collection methods to Table 1)

d.     The description made by Ishido S. and collaborators between lines 112 – 119 should be summarized within Table 4, and Table 3 should be move to supplementary data. This will help to put together the same type of information, and also to improve the results presentation.

e.     In table 4 could be also interesting to add the ratio of mutation type by patient. At this point of the work the authors already know that none of the patients were able to follow a clinical trial option. They should put this information on Supplementary table because it’s not relevant as a main information, and also remove all the misleading results where they mention the proportions of patients that could undergo clinical trials (for example: lines 191 – 193)

f.      In “Treatment after the comprehensive genomic profiling”, what the authors mention is interesting. However, more details about decision making and data analysis should be included, there is a concerning lack of information on how the statistical analysis or other methods were applied.

3.     Discussion

a.     Ishido S. et al make an excellent “state of the art” on how their findings are supporting or improving previously findings, however this part should be simplified and go to the point of the main findings of this work.

b.     The finding described between line 210 and 217 are not well described in results, they just appeared in this section and the authors called a main finding. The results need to be better work, and this will allow to focus the discussion (as mentioned above).

c.     The authors main a general claim (line 218-219), “Based on the results of this study, new molecular targeting agents for HCC, like the 218 β-catenin inhibitors and mTOR inhibitors would be promising options.”. How did the author male such a general claim when they just evaluated a cohort of 9 patients. Such claim should be focus on the studied population where they have a lot of advances in this kind of targeted therapy.

d.     As mentioned above, the author knows that none of the patients are going to clinical trials (line 238 - 240), they should not include this information as a main finding within results.

e.     Finally, the authors are fully aware of the limits of this study (line 244 – 249), thus they should be able to limit their conclusions and perspective to the studied population.

4.     Fgiures:

a.     Figure 2 should be improved and probably one more figure could be developed and added associated mainly to the main findings. 

Author Response

We sincerely thank you for reviewing our manuscript titled “Clinical Utility of Comprehensive Genomic Profiling in Patients with Unresectable Hepatocellular Carcinoma” and providing us the opportunity to resubmit our article to Cancers.

We have carefully addressed your comments and modified the manuscript in response to the extensive insights provided. Therefore, we respectfully request reconsideration of our manuscript. We have attached our responses to your comments with detailed, point-by-point explanations with the revised submission.

Reviewer 2 Report

The authors have performed Next-Gen sequencing analysis on unresectable Hepatocellular Carcinoma patients. The goal was to determine the ideal molecular targeted therapy for these patients.

1) In Fig. 1, instead of just mentioning the percentage, it would be helpful to also mention the number of patients having mutations in the respective genes.

2) The headings of figures 1 and 2 need to be adjusted to be in a consistent format. (Font type, size, position, etc.)

3) Figure 2 needs more explanation in terms of the abbreviations used in the figure. The legend for the figure can also be improved. Eg: the smoking status label and the yes/no label having the same color is sometimes confusing.

3) In Fig 2. There are a few typos for the word "Noting" which i guess was supposed to be "Nothing".

4) The method of Genomic sequencing analysis can be elaborated further. Information such as the following could be added:-

         a) What was the mutant allele frequency (mAF) of the mutations detected?, what thresholds were set to make the variant calling?

         b) What method was used to screen out all the VUS? Was the genomic DNA control samples from each patients analyzed separately?

5a) The outcome of subsequent treatments of individual patients is only briefly described in line 208. What about the rest of the patients? What were the outcomes of their subsequent treatments? Table 3 mentions that only 1 patient from this cohort underwent molecular targeted therapy. Was it after the genomic profiling or was it before? What was the outcome?

5b) A majority of the patients of this cohort have not undergone molecular targeted therapy after the genomic profiling. How then can there be conclusions drawn that genomic profiling is a beneficial clinical method to determine the ideal molecular targeted therapy?

Author Response

(The authors gave the same response as above.)

Round 2

Reviewer 1 Report

Ishido S. and collaborators investigated the clinical usefulness of comprehensive genomic profiling (CGP) in patients with unresectable hepatocellular carcinoma (uHCC) who received multiple systemic therapies in real-world practice. While Hepatocellular carcinoma (HCC) accounts for 75%–85% of primary liver cancer cases, the life expectancy of these patients has considerably improved with rapid advancement in systemic therapy. Recently, clinical trials have shown that median overall survival of advanced HCC was months, and only few biomarkers were used for decision-making. Moreover, national health insurance in Japan have been covering six systemic therapies, and sequencing of solid/liquid samples from patients with uHCC. This has led to a rapidly expanding number of individualized therapies that specifically target CGP in a patient's tumor. However, the clinical utility of these tools in uHCC remains unknown, for these reasons the authors aimed to investigate the clinical utility of CGP by in real-world practice by using FoundationOne®CDx (F1CDx) and FoundationOne®Liquid CDx (F1LCDx).

About the revised version some comments:

1.     Thanks to the authors for the further description of patients’ characteristics. Some comments about this: 

a.     Line 124 – 134, Hepatitis B and C viruses have been mentioned many times, it could be better to use the acronym for both HBV and HCV.

b.     The authors should probably need to simplify this paragraph; it is a bit confused how they try to introduce this new information.

2.     Thanks to authors for the better description of F1CDx and F1LCDx methodologies.

3.     Thanks to Ishido S. and collaborators for adding F1CDx and F1LCDx information on Table 1, however the description added to this table from line 151 to 158 should be simplified, and relevant information in this description should be added to the methods on F1CDx and F1LCDx.

4.     Line 164, there is an extra point on “by F1LCDx. (Table 1).”

5.     Thanks to the authors for taking into account the changes on the different tables.

6.     As mentioned before, Figure 2 still need improving, the quality of the figures isn’t visually attractive or transmit easily the message the authors try to share.

7.     Thanks once again to Ishido S. and collaborators for the additional information in “Treatment after the comprehensive genomic profiling”

8.     Thanks to the authors also for adding some important information also in the Discussion, which together with the additional information on methods and results, give a better idea of the important results and main goal of this paper.

Author Response

Thank you so much for your valuable comments. We sincerely appreciate that you spent your precious time and gave us a lot of meaningful suggestions. We have rewritten our manuscript in accordance with the reviewers' comments.

Reviewer 2 Report

My questions have been adequately addressed.

Author Response

Thank you for your kind comment. We sincerely appreciate that you spent your precious time.